# Coordination Polymers Based on Rhenium Octahedral Chalcocyanide Cluster Anions and Ag^+^ Cations with Bipyridine Analogs

**DOI:** 10.3390/molecules27227684

**Published:** 2022-11-09

**Authors:** Yulia M. Litvinova, Yakov M. Gayfulin, Taisiya S. Sukhikh, Konstantin A. Brylev, Yuri V. Mironov

**Affiliations:** Nikolaev Institute of Inorganic Chemistry SB RAS, Acad. Lavrentiev Ave., 3, 630090 Novosibirsk, Russia

**Keywords:** silver, coordination polymer, cyanometallate, cluster, rhenium, crystal structure

## Abstract

A series of six coordination polymers based on octahedral cluster anions [Re_6_Q_8_(CN)_6_]^4−^ (Q = S or Se) and Ag^+^ cations coordinated by bipyridine analogs were synthesized under solvothermal conditions. Their structures have been characterized by single crystal X-ray diffraction. Compounds **1** and **2** described by the general formula [{Ag(phen)}_4_Re_6_Q_8_(CN)_6_] (Q = Se (**1**), S (**2**); phen = 1,10-phenanthroline) exhibit layered structures assembled into a supramolecular network by CH…π contacts. At the same time, compounds [{Ag(bipym)}_2_Ag_2_Re_6_Se_8_(CN)_6_] (bipym = 2,2′-bipyrimidine) (**3**), [{Ag_2_(bipy)}Ag_2_Re_6_Se_8_(CN)_6_]·CH_3_CN (bipy = 4,4′-bipyridine) (**4**) and [{Ag(dpbp)}_4_Re_6_Q_8_(CN)_6_]·2H_2_O·2CH_3_CN (Q = Se (**5**), S (**6**); dpbp = 4,4′-Di(4-pyridyl)biphenyl)) evince framework structures. In **1**, **2**, **5** and **6** weak Ag⋯Ag interactions are observed. All the compounds show luminescence in the red region. The luminescence quantum yields and lifetimes were found to be notably higher than those for most of the coordination polymers based on the octahedral rhenium cluster complexes.

## 1. Introduction

Hexanuclear rhenium cluster complexes have been an object of intensive research for the last two decades [1,2]. Such cluster complexes have a number of attractive properties: they exhibit intense red photoluminescence with microsecond emission lifetimes and possess high thermal and hydrolytic stability [3,4,5]. Reversible oxidation of {Re_6_Q_8_}^2+^ (Q = S, Se or Te) cluster core accompanied by luminescence quenching makes it possible to “switch off” the emission of compounds in controllable manner and construct chemical sensors [6,7]. Chalcocyanide cluster complexes [Re_6_Q_8_(CN)_6_]^4−^ (Q = S, Se, Te) have all these fascinating properties and additionally possess six cyanide ligands, which are capable of forming covalent bonds with 3*d* and 4*f* metal cations [8,9,10,11]. The current trend in the chemistry of coordination polymers is directed to the design of compounds with interesting multifunctional properties, and the rhenium chalcocyanide clusters make very suitable building blocks for such purpose. It was shown earlier that coordination polymers (CPs) built on [Re_6_Q_8_(CN)_6_]^4−^ anions transition metal cations and demonstrate weak emission with low quantum yields relative to simple salts of [Re_6_Q_8_(CN)_6_]^4−^ with alkali metal or organic cations [12]. Therefore, the improvement of photophysical characteristics of CPs based on [Re_6_Q_8_(CN)_6_]^4−^ can enhance their potential applicability as multifunctional materials. The combination of properties inherent for different building blocks and nodes is one of the common approaches for constructing compounds with predetermined properties.

In this work, Ag^+^ cations with the 2,2′-bipyridine, 4,4′-di- and 4,4′-bipyridine derivatives were introduced into the structure of CPs based on the [Re_6_Q_8_(CN)_6_]^4−^ (Q = S, Se) anionic complexes. Compounds based on Ag^+^ ions are of interest due to electrical conductivity [13], biological activity [14] and, especially, emissive properties [15,16,17]. Diversity of coordination geometry of silver complexes makes such cations convenient nodes for constructing CPs with rich architectural varieties. In addition, the ability of Ag^+^ to form metal–metal interactions increases structural diversity and also influences the emission properties [18,19,20]. Despite the fact that Ag^+^-containing compounds have been known for a long time, their investigation was hampered for a long time due to low photostability. The number of compounds based on Ag^+^ cations and organic ligands had arisen drastically in the last two decades. Partially, it is associated with a great interest in a group of compounds exhibiting thermally activated delayed fluorescence (TADF). It is assumed that such compounds are promising for the new generation of OLEDS [21]. Silver, as a soft Lewis acid, has a good affinity to nitrogen. N-donor aromatic organic molecules are classical ligands for the synthesis of compounds based on Ag^+^ cations [22]. It was shown recently that compounds based on the cluster anions [Re_6_Q_8_(CN)_6_]^4−^ (Q = S or Se) and [Ag(2,2′-bipy)]^+^ cations possessed spectral characteristics surpassing the characteristics of the initial cluster salts [23]. In particular, investigation of luminescence properties showed an increase in the quantum yield and lifetime of the obtained CPs. In this work, we describe the syntheses and characterization of six new coordination polymers based on the cluster anions [Re_6_Q_8_(CN)_6_]^4−^ (Q = S or Se) and Ag^+^ complexes with bipyridine analogs. Structures, composition, spectroscopic and photophysical properties of the new compounds were investigated. A great effect of coordination of the Ag^+^ complexes on emission parameters of cluster anions was revealed.

## 2. Experimental Section

### 2.1. Materials and Methods

The starting cluster salts K_4_[Re_6_Se_8_(CN)_6_]·3.5H_2_O and Cs_3_K[Re_6_S_8_(CN)_6_]·2H_2_O were prepared as described [24,25]. Other reagents were used as purchased. Elemental analysis was made on a EuroVector EA3000 analyzer (EuroVector, Pavia, Italy). IR spectra in KBr pellets in the range 4000–375 cm^−1^ were recorded on a Bruker Scimitar FTS 2000 spectrometer (Bruker Corporation, Billerica, MA, USA). Energy dispersive spectroscopy (EDS) was performed on a Hitachi TM-3000 electron microscope equipped with a Bruker Nano EDS analyzer (Hitachi, Ltd., Chiyoda City, Tokyo, Japan). Powder X-ray diffraction (PXRD) was performed at room temperature on a Shimadzu XRD-7000 (Shinadzu, Kyoto, Japan) diffractometer (Cu Kα radiation, graphite monochromator). Room temperature excitation and emission spectra of powdered samples were recorded with a Horiba Jobin Yvon Fluorolog 3 photoluminescence spectrometer equipped with 450 W ozone-free Xe-lamp (Horiba, Ltd., Kyoto, Japan), cooled PC177CE-010 photon detection module (Hamamatsu Photonics K.K., Hamamatsu City, Shizuoka, Japan) with a PMT R2658 and double grating excitation and emission monochromators (Products for Research, Inc., Danvers, MA, USA). Excitation and emission spectra were corrected for source intensity (lamp and grating) and emission spectral response (detector and grating) by standard correction curves.

#### Single Crystal Diffraction Studies

Diffraction data for single crystals of **1**–**4** and **6** were collected with a Bruker Apex DUO (Bruker Corporation, Billerica, MA, USA) diffractometer equipped with a 4 K CCD area detector using the graphite-monochromated MoKα radiation (λ = 0.71073 Å) at 150 K. The φ- and ω-scan techniques were employed to measure intensities. Absorption corrections were applied with the use of the SADABS program [26]. The crystal structure was solved using the SHELXT [27] and were refined using SHELXL [28]. Diffraction data for single crystal **5** were obtained on an Agilent Xcalibur diffractometer equipped with a CCD AtlasS2 detector (MoKα, graphite monochromator, ω scans) (Agilent Technologies, Santa Clara, CA, USA). Integration, absorption correction and determination of unit cell parameters were performed using the CrysAlisPro program package [29]. Positions of hydrogen atoms of organic ligands were calculated geometrically and refined in the riding model. The crystallographic data and details of the structure refinements are summarized in Appendix A. Selected bond distances are shown in Appendix A. CCDC 2212912–2212917 contain the crystallographic data for compounds **1**–**6**, respectively. These data can be obtained free of charge from The Cambridge Crystallographic Data Centre via www.ccdc.cam.ac.uk/structures (accessed on 14 October 2022)

### 2.2. Syntheses

Synthesis of [{Ag(phen)}_4_Re_6_Se_8_(CN)_6_] (**1**). 1,10-phenanthroline (15 mg, 0.083 mmol) was dissolved in 2 mL of EtOH. KAg(CN)_2_ (10 mg, 0.050 mmol) was dissolved in 1 mL of distilled H_2_O. Solution of K_4_[Re_6_Se_8_(CN)_6_]·3.5H_2_O (25 mg, 0.011 mmol) in 1 mL of H_2_O was placed in a glass ampoule, then K[Ag(CN)_2_] and phen solutions were sequentially added. The ampoule was sealed and heated at 140 °C for 48 h. Rhomboid crystals were formed on the bottom of the ampoule. Yield: 21 mg (63% based on [Re_6_S_8_(CN)_6_]^4−^ anion). FT-IR (ν_max_, cm^−1^): ν(CN)—2133, 2121; phen—3047, 1620, 1585, 1568, 1508, 1423, 1338, 1256, 1219, 1136, 1097, 840, 769, 729, 630. Anal. calcd (%) for C_54_H_32_Ag_4_N_14_Re_6_Se_8_: C, 21.21; H, 1.05; N, 6.41. Found: C, 21.50; H, 1.10; N, 6.50. EDS: Ag:Re:Se = 3.8:6.0:8.2.

Synthesis of [{Ag(phen)}_4_Re_6_S_8_(CN)_6_] (**2**). Compound **2** was synthesized using the same procedure as for **1** except using Cs_3_K[Re_6_S_8_(CN)_6_]·2H_2_O (25 mg, 0.011 mmol). Yield: 11 mg (37% based on [Re_6_S_8_(CN)_6_]^4−^ anion). FT-IR (ν_max_, cm^−1^): ν(CN)—2140, 2117; phen—3049, 1618, 1587, 1558, 1508, 1421, 1338, 1219, 1137, 1095, 842, 729, 630. Anal. calcd (%) for C_54_H_32_Ag_4_N_14_Re_6_S_8_: C, 24.18; H, 1.20; N, 7.31; S, 9.56. Found: C, 24.30; H, 1.20; N, 7.40; S, 9.70. EDS: Ag:Re:S = 3.8:6.0:8.2.

Synthesis of [{Ag(bipym)}_2_Ag_2_Re_6_Se_8_(CN)_6_] (**3**). 2,2′-bipyrimidine (10 mg, 0.163 mmol) was dissolved in 2 mL of EtOH. K[Ag(CN)_2_] (10 mg, 0.050 mmol) was dissolved in 1 mL of distilled H_2_O. Solution of K_4_[Re_6_Se_8_(CN)_6_]·3.5H_2_O (25 mg, 0.011 mmol) in 1 mL of H_2_O was placed in a glass ampoule, then K[Ag(CN)_2_] and 2,2′-bipyrimidine solutions were sequentially added. The ampoule was sealed and heated at 140 °C for 40 h. Several red rhomboid crystals were formed on the bottom of the ampoule mixed with crystalline white powder and amorphous gray powder. A crystal for single-crystal X-Ray diffraction studies was selected manually. EDS: Ag:Re:Se = 4.1:6.0:8.1.

Synthesis of [{Ag_2_(bipy)}Ag_2_Re_6_Se_8_(CN)_6_]·CH_3_CN (**4**). 4,4′-bipyridine (20 mg, 0.128 mmol) was dissolved in 1 mL of EtOH. KAg(CN)_2_ (10 mg, 0.050 mmol) was dissolved in 1 mL of distilled H_2_O. Solution of K_4_[Re_6_Se_8_(CN)_6_]·3.5H_2_O (20 mg, 0.011 mmol) in 1 mL of H_2_O was placed in a glass ampoule, then K[Ag(CN)_2_] and 4,4′-bipydine solutions as well as 1 mL of CH_3_CN were sequentially added. Resulted solution turned cloudy. The ampoule was sealed and heated at 140 °C for 72 h. As a result, the solution became clear. Then ampoule was left in the dark place for 240 h. Several needle-like crystals were formed mixed with crystalline white powder and amorphous black powder. Crystal for single-crystal X-Ray diffraction studies was selected manually. EDS: Ag:Re:Se = 3.9:6.0:8.2.

Synthesis of [{Ag(dpbp)}_4_Re_6_Se_8_(CN)_6_] 2H_2_O·2CH_3_CN (**5**). For the synthesis of the compound, two ampoules sealed together with a common neck were used (Figure 1). 4,4′-Di(4-pyridyl)biphenyl (bpbp) (25 mg, 0.081 mmol) was placed in one ampoule together with solutions of K_4_[Re_6_Se_8_(CN)_6_]·3.5H_2_O (25 mg, 0.011 mmol) in 1 mL of H_2_O and K[Ag(CN)_2_] (10 mg, 0.050 mmol) in 1 mL of H_2_O. Additionally, 1 mL of EtOH and 1 mL of CH_3_CN were added to the ampoule. The neck was then sealed and the double ampoule was left in the furnace at 150 °C. After 2 weeks, flat square crystals formed at the bottom of the ampoule with reagents. The hot mother solution was carefully poured into the second ampoule resulting in the pure crystalline product **5**. Crystals were washed with H_2_O and dried on the air. Yield: 14 mg (35% based on the [Re_6_Se_8_(CN)_6_]^4−^ anion). FT-IR (ν_max_, cm^−1^): ν(CN)—20112, 2090; dpbp—1604, 1558, 1541, 1508, 1473, 1458, 1419, 1396, 1261, 1068, 1004, 804, 756, 669, 557. Anal. calcd (%) for C_94_H_64_Ag_4_N_14_Re_6_Se_8_: C, 31.91; H, 2.02; N, 6.07. Found: C, 32.1; H, 2.10; N, 6.00. EDS: Ag:Re:Se = 4.0:6.0:8.1.

Synthesis of [{Ag(dpbp)}_4_Re_6_S_8_(CN)_6_]·2H_2_O·2CH_3_CN (**6**). Compound **6** was synthesized using the same procedure as for **5** except using Cs_3_K[Re_6_S_8_(CN)_6_]·2H_2_O (25 mg, 0.011 mmol) as starting cluster salt. Yield: 5 mg (14% based on [Re_6_S_8_(CN)_6_]^4−^ anion). FT-IR (ν_max_, cm^−1^): ν(CN)—2113, 2090; dpbp—1604, 1558, 1541, 1506, 1481, 1458, 1417, 1396, 1338, 1224, 1066, 1004, 804, 756, 669, 559. Anal. calcd (%) for C_94_H_64_Ag_4_N_14_Re_6_S_8_: C, 35.53; H, 2.25; N, 6.76; S, 7.74. Found: C, 35.80; H, 2.30; N, 6.50; S, 7.70. EDS: Ag:Re:Se = 3.9:6.0:8.2.

## 3. Results and Discussion

### 3.1. Synthesis

Relatively few compounds based on cyanometallates and silver cations are known. There are even fewer examples when [Ag(CN)_2_]^−^ acts as a source of silver cations in such syntheses [30,31]. In the case of octahedral cluster cyanide complexes, the choice of the silver source was found to be critical for the preparation of crystalline coordination polymers with organic ligands. The coordination polymers **1**–**6** were obtained by a solvothermal synthesis as a result of the interaction of [Ag(CN)_2_]^−^, [Re_6_Q_8_(CN)_6_]^4−^ (Q = S or Se) and corresponding organic ligands in water/organic solvent mixtures. It should be noted that, according to the results of IR spectroscopy, all CN-groups in the obtained compounds are equivalent. The frequency of ν(CN) vibration is 2125–2155 cm^−1^ matching well the corresponding frequencies in other coordination polymers based on [Re_6_Q_8_(CN)_6_]^4−^ (Q = S, Se) cluster anions [7,32]. It can be assumed that the stage of destruction of the [Ag(CN)_2_]^−^ anion is the limiting one and determines the reaction rate, while the coordination of Ag^+^ cations to the cluster complex occurs very quickly. Indeed, the introduction of simple salts, such as silver nitrate, instead of K[Ag(CN)_2_] into the reaction mixtures led to immediate formation of amorphous products, which, according to elemental analysis, did not contain organic ligands. We propose that use of [Ag(CN)_2_]^−^ increases the ability of organic ligands to compete for silver coordination sites with the cyano-groups of the cluster complexes.

### 3.2. Crystal Structures

Compound **1** crystallizes as rhomboidal plates in the triclinic space group *P*1. The asymmetric units of compound **1** (Figure 2) contain half of [Re_6_Se_8_(CN)_6_]^4−^ cluster anion, two Ag^+^ ions and two phenanthroline ligands. The geometric parameters of [Re_6_Se_8_(CN)_6_]^4−^ cluster anion are typical and correlate well with the literature data (Appendix A). The coordination environment of Ag1 atom includes two nitrogen atoms of the CN-groups and two N atoms of the phenanthroline molecule with the formation of distorted square planar geometry (Figure 3a). Ag1–N1 distances (between Ag1 atom and two N atoms of cyanide groups of adjacent cluster complexes) are 2.26 and 2.67 Å. The last distance is longer probably due to the bridge coordination of N atom of cyanide group (Figure 4). Ag1–N101 and Ag1–N102 distances are equal to 2.32 and 2.44 Å, respectively. The environment of Ag1 is completed by another Ag1 atom located at a distance of 3.25 Å (Figure 4), which corresponds to a weak argentophilic interaction [17]. The coordination environment of Ag2 atom includes two N atoms of the cyano-groups and two N atoms of the phenanthroline molecule. The coordination polyhedron is a distorted tetrahedron (Figure 3b). Distances between phen N201 and N202 atoms and Ag2 atom are equal to 2.32 and 2.39 Å; cyanide groups form Ag2–N2 and Ag2–N3 bonds with lengths of 2.37 and 2.14 Å, respectively. In the structure, cluster complexes are linked in a chain via four Ag1 atoms pairwise connected by N atoms of bridged cyanide l (Figure 4). The chains extend parallel to each other and are linked by Ag2 atoms ([Ag(phen)]^+^ complexes) bonded to the clusters cyanide groups of adjacent chains (Figure 5). A number of CH…π interactions between phenanthroline aromatic rings generate frameworks without an empty space accessible for solvate molecules (Figure 6).

Compound **2** is isostructural to compound **1**. The main geometric characteristics of compound **2** are presented in Appendix A.

1,10-phenanthroline is a common chelating N-donor ligand for 3*d* and 4*f* metals. Silver-based compounds are no exception. More than 200 compounds based on silver cations and phenanthroline or its derivatives with various additional ligands are presented in the literature [33,34,35]. At the same time, there are only a few examples of compounds with 1,10-phenanthroline that have argentophilic interactions. Among them, a number of compounds can be distinguished where the formation of argentophilic interactions was facilitated by π–π interactions between phenanthroline molecules of neighboring cationic complexes [Ag(phen)*_n_*]^+^ [36,37,38,39,40,41]. It could be expected that the formation of π-stacking between phenanthroline molecules coordinated to Ag^+^ ions would favor the formation of argentophilic interactions in the structures of compounds **1** and **2**. However, in the dimeric cationic complex (phen)Ag⋯Ag(phen), the phenanthroline molecules lie in the same plane, which makes it impossible for stacking to form between them (Figure 5). Thus, phenanthroline acts as a bulky ligand that blocks some of the coordination sites of the silver atom [42,43,44], and the formation of argentophilic interactions may occur due to packing effects. The formation of the argentophilic interactions in compounds **1** and **2** could also be facilitated by μ mode of N donor atom of cyanide group.

Compound **3** crystallizes in orthorhombic space group *Pbca*. The asymmetric unit contains half of [Re_6_Se_8_(CN)_6_]^4−^ cluster anion, one Ag^+^ cation disordered over two close positions (Ag1 and Ag2 with 0.85 and 0.15 occupancies, respectively), one Ag^+^ with the full occupancy, four N and eight C atoms of 2,2′-bypirimidine (bipym) ligand (Figure 7). Coordination environment of Ag^+^ in the Ag1 position includes two N atoms (N11 and N12) of the bipym molecule and two N atoms (N1 and N3) of cyanide groups. Ag1–N1 and Ag1–N3 distances are equal to 2.33 and 2.21 Å, while Ag1–N11 and Ag1–N12 distances—2.46 and 2.49 Å, respectively. Coordination polyhedron is a distorted tetrahedron (Figure 8a). Coordination environment of Ag2 is represented by two N atoms of two cyanide groups and two N atoms of bipym. Ag2–N1 and Ag2–N3 distances are equal to 2.43 and 1.99 Å. It should be noted that the Ag2–N_bipym_ distances are 2.137 and 2.731 Å, which indicates a weak contact in the second case. With this elongated Ag2–N bond taken into account, the coordination number of Ag2 is 4, and the coordination polyhedron is a strongly distorted tetrahedron (Figure 8b). Coordination environment of Ag3 atom is presented by two N atoms (N21 and N22) of bipym ligand, two Se atoms, one N (N1) and one C atoms of cyanide ligands. The bond lengths of Ag3–N1, Ag3–N21 and Ag3–N22 are 2.31, 2.39 and 2.56 Å, respectively. The Ag–Se bond lengths are 2.87 and 3.04 Å implying the formation of relatively strong chalcophilic interactions. Ag3–C2 bond length is equal to 2.49 Å. Coordination polyhedron around Ag3 is trigonal prism (Figure 8c). Coordination environment of the cluster complex includes 8 Ag atoms (Figure 9). Two of the cyanide groups act as μ-N-donor ligands binding Ag1 and Ag3 atoms; two cyanide groups bind one Ag1 atom. Additionally, two Ag3 atoms form bonds with two Se atoms and C atom (Figure 9). Thus, a pseudocubic crystal structure is formed (Figure 10). Additionally, Ag1 and Ag3 atoms linked via bipym molecule. It is noteworthy that there is no π-stacking between aromatic rings or argentophilic interactions in the structure.

The number of reported compounds based on Ag^+^ cations and 2,2′-bipyrimidine is rather small. Without the inclusion of additional ligands, compounds of such type have chain (1D) polymeric structures, and in the chains bipym ligand link two metal centers [45,46,47,48]. The introduction of additional bridging ligands, such as oxalate, can also lead to the formation of 1D structures [45,48]. Small clusters, like [Cp_2_Mo_2_(CO)_4_(η_2_-P_2_)], can act as a coligand resulting in the formation of layered structures [49]: 2,2′-bipyrimidine links silver atoms to form chains, and the chains are connected into layers by the organometallic complex [Cp_2_Mo_2_(CO)_4_(η_2_-P_2_)]. For one compound, described in [49], argentophilic interactions are mentioned. It should be noted that this is the only example of a coordination polymer based on silver cations and 2,2′-bipyrimidine, in which the presence of argentophilic interactions was revealed. It should also be noted that there are no framework compounds among coordination polymers built from Ag^+^ and bipym.

Compound **4** crystallizes in monoclinic crystal system, *C*2/*m* space group. The asymmetric unit is presented by two Re atoms, three Se atoms, two N and C atoms of cyanide groups, two Ag atom and one ring (five C, one N and four H atoms) of 4,4′-bypiridine (Figure 11). Besides, one solvate molecule of acetonitrile is presented. The coordination environment of Ag1 atom includes two N11 atoms of bipy and one N2 atom of cyanide group with bond length equal to 2.23 and 2.47 Å, respectively. Additionally, Ag1 forms a long bond with the N2 atom of the adjacent cluster complex with length equal to 2.68 Å. The coordination polyhedron is a distorted tetrahedron (Figure 12a). The coordination environment of Ag2 atom consists of two N atoms of cyanide groups (*d*_Ag–N_ = 2.14 Å) and one Se atom (*d*_Ag–Se_ = 2.78 Å). Moreover, the Ag2 atom forms a bond with the C atom of cyanide group of the neighbouring cluster complex with *d*_Ag–C_ = 2.50 Å. The coordination polyhedron is a distorted tetrahedron (Figure 12b). The cluster complex bonds six Ag atoms via cyanide groups. Two Ag1 atoms are apical, and four Ag2 atoms are bridging. Ag1 atoms are linked via bipy ligands. Thus, a polymeric ribbon (1D structure) is formed (Figure 13). Each cluster complex forms two Ag–Se bonds with two Ag atoms of the neighboring ribbons. Additionally, Ag1 and Ag2 atoms form bonds with N1 and C1 atoms of the neighboring ribbons forming a framework (Figure 14). It should be noted that there are no π–π interactions between aromatic rings of 4,4′-bipyridine molecules or argentophilic interactions in the structure **4**.

4,4′-bipyridine is a common ligand for the synthesis of compounds based on 3*d* metals.

Usually, the binding of silver cations by bipyridine molecules results in the formation of polymeric chains [Ag(bipy)]^+^*_n_*, i.e., 1D structures, which have linear or zigzag shapes depending on the coordination environment of Ag^+^. There are numerous compounds in the literature that include such moieties. To increase the dimensionality of a structure, various polydentate ligands are introduced. Presence of aromatic rings of the ligand forming π–π interactions often promote formation of argentophilic interactions in such structures [50,51]. The inclusion of bidentate ligands may not lead to increase of the dimensionality of the coordination polymer. The ligand linking two [Ag(bipy)]^+^*_n_* chains through silver atoms forms one-dimensional ladder motifs, which are further bonded into a layer through π–π interactions between bipyridyl aromatic rings and argentophilic interactions [52,53,54]. In addition, bidentate ligands can coordinate two silver atoms of [Ag(bipy)]^+^*_n_* chains located parallel to each other and linked by π–π interactions, stabilizing argentophilic interactions without leading to an increase in the dimension of the coordination polymer [55]. Bulky, rigid and long linkers make it possible to obtain layers, which form a supramolecular framework due to the presence of π–π interactions between bipyridine molecules [54,56]. At the same time, argentophilic interactions can be formed between the silver atoms of neighboring layers, which also stabilize the packing [57]. The number of framework (3D) compounds based on silver cation and 4,4′-bipyridine is small. For example, a framework compound was obtained by introducing 3,5-(di(2′,5′-dicarboxylphenyl)benozoic acid, where each ligand coordinates four silver atoms [58]. It is also worth highlighting a number of works on the synthesis of compounds based on polyoxometalates (POMs) and silver cations. Among them, there are compounds where POMs acted as a coligand, binding [Ag(bipy)]^+^*_n_* chain motifs. For example, in the work of H. Yang et al., [PW_12_O_40_] acts as an octadentate ligand, while linking six [Ag(bipy)]^+^*_n_* chains through oxygen atoms into a framework [59]. Similarly, such POMs as [Mo_5_P_2_O_23_], [SiW_12_O_40_] and [GeW_12_O_40_], acting as hexadentate ligands, coordinate the silver atoms of the [Ag(bipy)]^+^*_n_* chains to form 3D structures [60,61]. In comparison with such POMs as [SiW_12_O_40_] and [GeW_12_O_40_], the cluster complex is much smaller, which could explain the binding of only two [Ag(bipy)]^+^*_n_* chains. It is noteworthy that [Mo_5_P_2_O_23_] has comparable size with the [Re_6_Q_8_(CN)_6_]^4−^ cluster. However, all six [Ag(bipy)]^+^*_n_* chains linked to [Mo_5_P_2_O_23_] in such a way that their mutual arrangement is almost parallel to each other, which is sterically impossible in the case of the hexarhenium cluster complex.

The asymmetric unit of compound **5** contains three Re atoms of the cluster core, three cyanide ligands, four μ_3_-Se ligands, two Ag^+^ cations and two dpbp ligands (Figure 15). The coordination environment of Ag1 includes two N atoms of two different dpbp ligands (N11 and N12), one N atom of a cyanide group (N1) and one Se atom. Ag1–N11, Ag–N12 and Ag1–N1 bond lengths are equal to 2.2, 2.21 and 2.50 Å, respectively. The Ag1–Se1 bond length is equal to 3.06 Å. Ag1 is tetra-coordinated with the geometry of coordination environment close to square planar (Figure 16a). The coordination environment of Ag2 atom includes two N atoms of two dpbp ligands and one N atom of a cyanide group. The Ag–N distances are equal to 2.177 and 2.164 Å in the case of dpbp ligands and 2.62 Å for CN group. Three coordinated Ag2 with N_dpbp_–Ag–N_dpbp_ angles equal 176.6° and have a T-shaped environment (Figure 16b). The coordination environment of the cluster complex is presented by four Ag atoms (two Ag1 and two Ag2) bonded to cyanide groups. Silver atoms, in turn, are linked in chains by N atoms of dpbp ligand. Thus, four [AgL]^+^*_n_* polymeric chains linked by the cluster complexes form a one-dimensional [(Ag(dpbp))_4_[Re_6_Se_8_(CN)_6_]*_n_* fragment (Figure 17 and Figure 18). Such chains are bound into a framework due to numerous Ag–Se bonds (Figure 19). Each cluster complex provides two selenium atoms for bonding with two silver atoms of two neighboring chains. There are very weak Ag⋯Ag interactions in the structure. The distance between two Ag2 atoms is 3.29 Å, which is close to the sum of van der Waals radii of 3.44 Å.

4,4′-Di(4-pyridyl)biphenyl is not a common ligand for the synthesis of coordination compounds. In the literature, there are about 30 examples of coordination polymers based on this ligand and 3*d* metal cations. Moreover, among them there are no examples of compounds with the metals of group 11 except two examples of compounds with copper cations [62,63]. The conditions for the synthesis of compounds **4**–**6** were similar. However, compound **4** formed after a long stay of the sealed ampoule in the dark after heating, while the crystalline product for compounds **5** and **6** began to form after 48 h in the furnace. The structures of compounds **4** and **5** also differ significantly. In both cases, linear chains [Ag(L)]^+^*_n_* were formed. However, in the case of compound **5**, the cluster complex coordinated four such chains, while for compound **2**, the coordination environment of the cluster complex included only two chains. In addition, as mentioned above, compound **4** had additional bridging silver atoms that did not include bipy molecules in their coordination environment. Moreover, the absence of π–π interactions for compound **5** is of interest. Each cluster complex in both compounds provided two selenium atoms for bonding with silver atoms.

### 3.3. Luminescence Properties

Luminescence properties of chalcocyanide complexes [{Re_6_Q_8_}(CN)_6_]^4−^ (Q = S, Se or Te) were studied in solution and in the solid state. The tellurocyanide cluster complex shows negligible luminescence while sulfido- and selenocyanide clusters brightly emit in visible and near-infrared regions upon ultraviolet light excitation [3]. It was shown that some coordination polymers based on cluster anions [Re_6_Q_8_(CN)_6_]^4−^ and transition metal cations possess luminescence properties typical for the chalcocyanide cluster complexes [12,23,64,65]. It is noteworthy that, in the luminescence spectra the coordination polymers based on the cluster anions and lanthanide cations usually only a broad band of the corresponding cyanide cluster complex is presented without lanthanides emission bands, and the luminescence is often much weaker than that of starting alkali metal salts of the clusters [7,64]. Two coordination polymers, namely [{Ag(bpy)}{Ag_4_(bpy)_4_(µ-CN)}{Re_6_Q_8_(CN)_6_}] (Q = S or S), were the first [Re_6_Q_8_(CN)_6_]^4−^-based coordination polymers for which luminescence characteristics comparable with the parent cluster salts were reported [21]. Emission spectra of the compounds **1**, **2**, **5** and **6** show the emission band characteristic of [Re_6_Q_8_(CN)_6_]^4−^ clusters without noticeable contribution from the Ag^+^ complex with ligand (Figure 20). Emission lifetimes and quantum yields for compounds **5** and **6** are comparable with parent compounds (Bu_4_N)_4_[Re_6_Q_8_(CN)_6_] (Q = S, Se). But compounds **1** and **2** possess more impressive photophysical characteristics (Table 1).

Complexes of Ag^+^ are known for their luminescence properties. N-donor conjugated molecules bonded to Ag^+^ cation increase their conformational rigidity leading to the reduction in energy loss by intramolecular vibrational and rotational motions and, as a result, to enhanced luminescence [22]. Moreover, the nature of the luminescence is determined by the ligand structure and the structure of obtained metal complex. Emission bands of free 1,10-phen and dpbp (Figure 21 and Figure 22) are located in the wavelength region 350–500 nm and overlap with excitation spectra of the chalcocyanide cluster complexes, which cover the wavelength region from 210 to more than 500 nm. Thus, there is every reason to assert that intense and long-lived emission of compounds **1**, **2**, **5** and **6** can be attributed to the Förster Resonance Energy Transfer (FRET) mechanism with the Ag^+^ complex acting as donor and cluster anion as the acceptor because of the mentioned overlap of their emission and excitation spectra.

## 4. Conclusions

In conclusion, we have obtained 6 new coordination compounds based on Ag^+^ cations and [Re_6_Q_8_(CN)_6_]^4−^ cluster anions (Q = S or Se) with various N- heterocyclic aromatic ligands. The resulting compounds have both 2D and 3D structures. Investigation of luminescence properties of the new compounds showed they all demonstrate a broad emission band of the corresponding chalcocyanide {Re_6_} cluster. Photophysical characteristics of compounds **5** and **6** are comparable with parent compounds (Bu_4_N)_4_[Re_6_Q_8_(CN)_6_] (Q = S, Se). Emission lifetimes and quantum yields of **1** and **2** are much higher than those of the parent compounds. In addition, these compounds exhibit the highest photophysical characteristics among coordination polymers based on the cluster complexes [Re_6_Q_8_(CN)_6_]^4−^ with 3*d* and 4*f* metals. This work has shown that the use of Ag^+^ cations coordinated by heterocyclic N-donor ligands as components of coordination polymers based on rhenium cluster complexes makes it possible to obtain polymeric materials with bright cluster-based red emission. Most importantly, the emission lifetimes and quantum yields for the polymers in some cases exceed those for discrete cluster ions. Further development of this chemistry may result in obtaining new luminescent coordination polymers, which are potentially applicable as multifunctional materials.

## Figures and Tables

**Figure 1 molecules-27-07684-f001:**
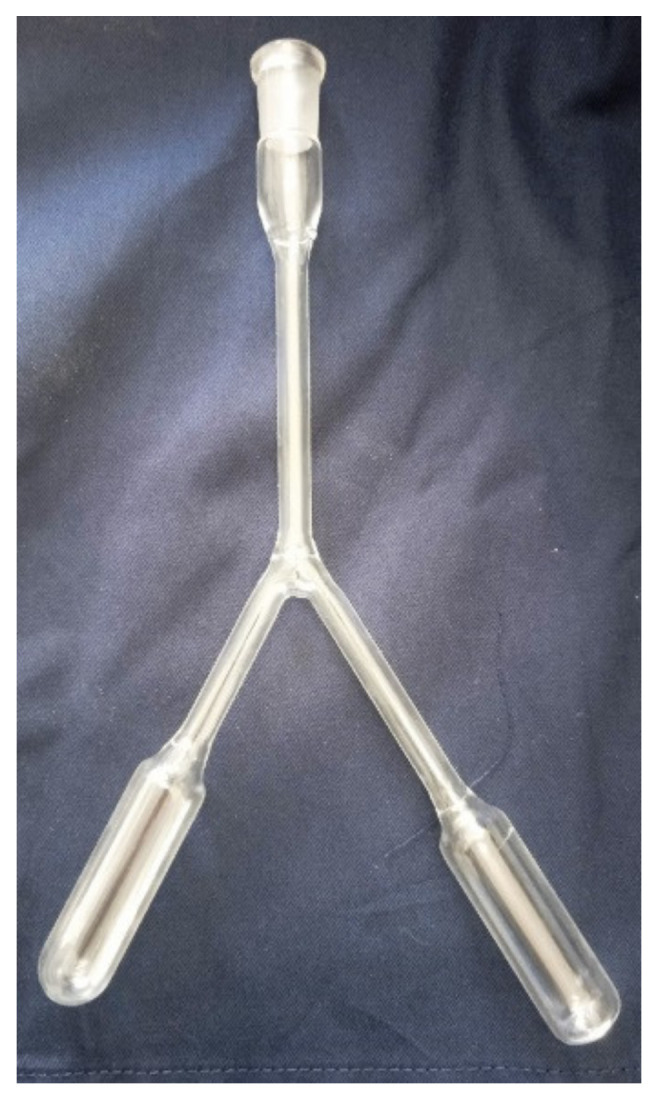
Double ampoule for the synthesis of the compounds **5** and **6**.

**Figure 2 molecules-27-07684-f002:**
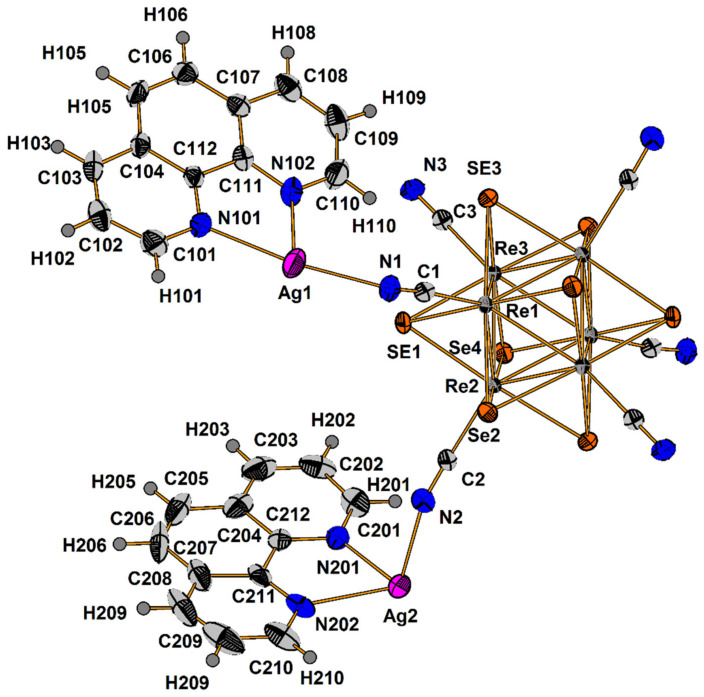
ORTEP drawing of the asymmetric unit in the structure of **1**. Thermal ellipsoids of 75% probability.

**Figure 3 molecules-27-07684-f003:**
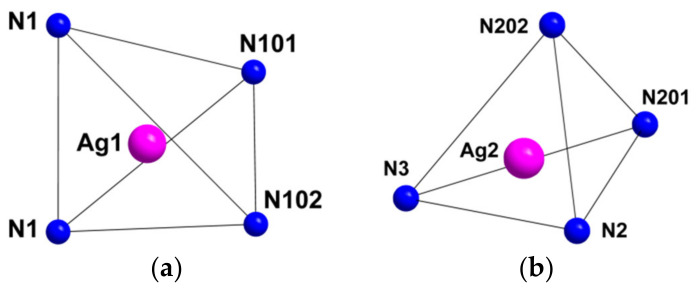
Coordination polyhedra of Ag1 (**a**) and Ag2 (**b**) atoms.

**Figure 4 molecules-27-07684-f004:**
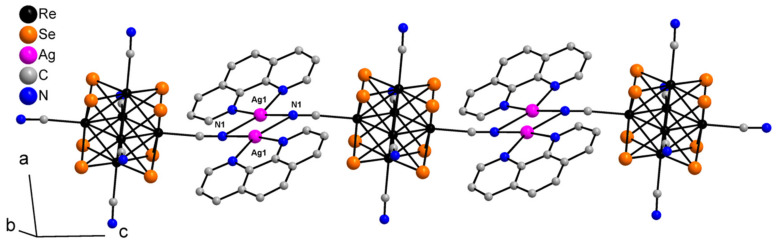
Fragment of compound **1**.

**Figure 5 molecules-27-07684-f005:**
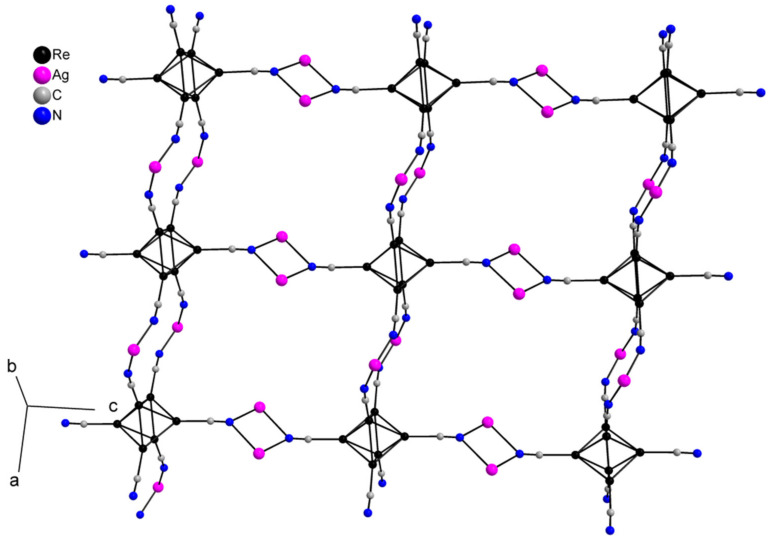
A layer structure of compound **1**. Se atoms and phen molecules are omitted for clarity.

**Figure 6 molecules-27-07684-f006:**
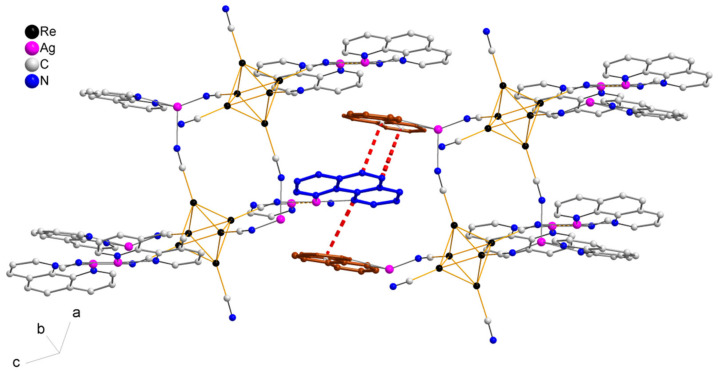
A scheme of CH…π interactions between phenanthroline aromatic rings.

**Figure 7 molecules-27-07684-f007:**
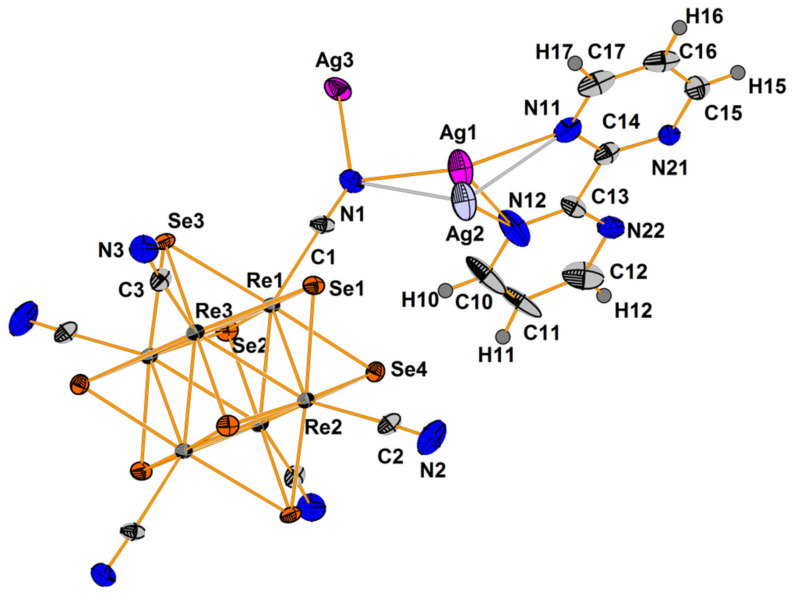
ORTEP drawing of the asymmetric unit in the structure of **3**. Thermal ellipsoids of 75% probability.

**Figure 8 molecules-27-07684-f008:**
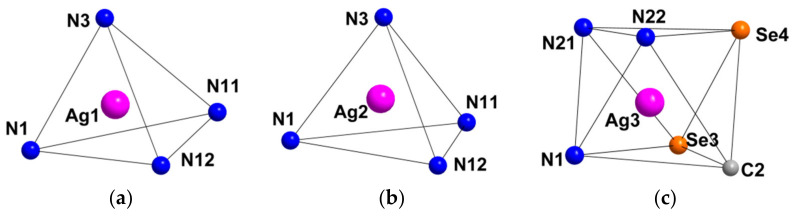
Coordination polyhedra of Ag1 (**a**), Ag2 (**b**) and Ag3 (**c**) atoms.

**Figure 9 molecules-27-07684-f009:**
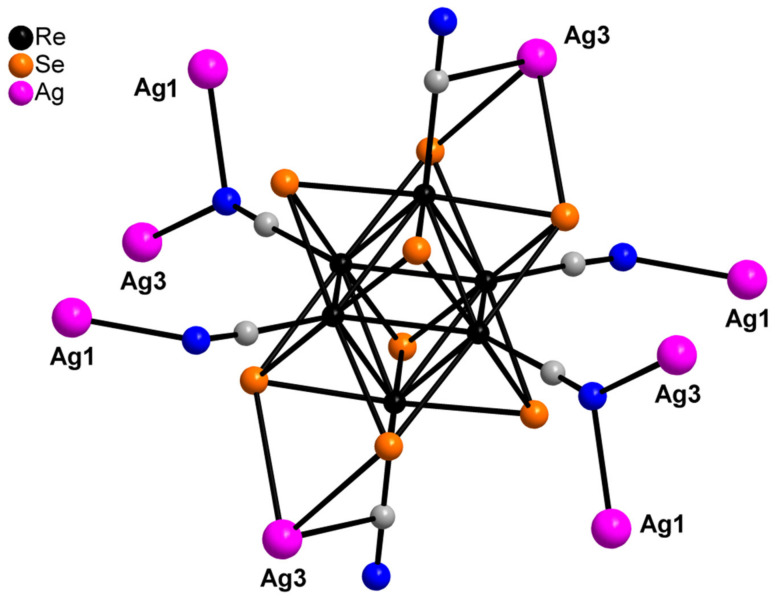
Coordination environment of cluster complex in the structure of compound **3**.

**Figure 10 molecules-27-07684-f010:**
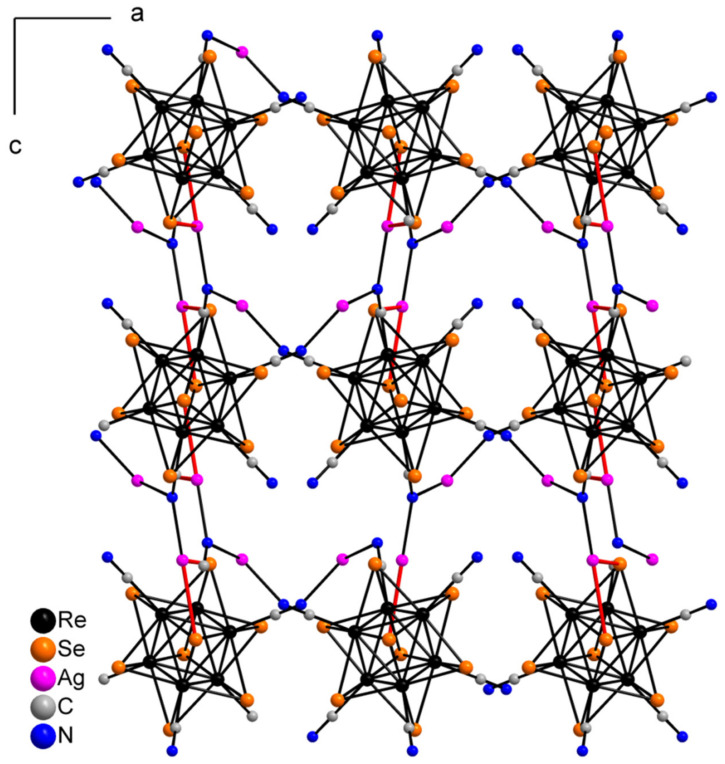
A fragment of framework in **3**. bipym ligands are omitted for clarity.

**Figure 11 molecules-27-07684-f011:**
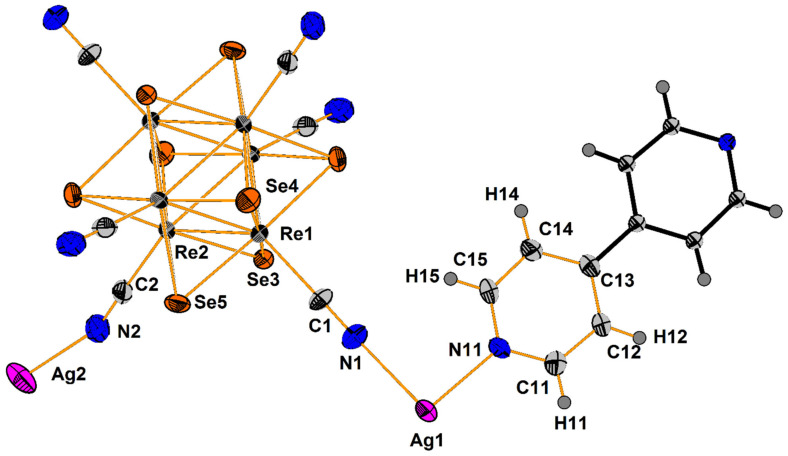
ORTEP drawing of the asymmetric unit in the structure of **4**. Thermal ellipsoids of 75% probability.

**Figure 12 molecules-27-07684-f012:**
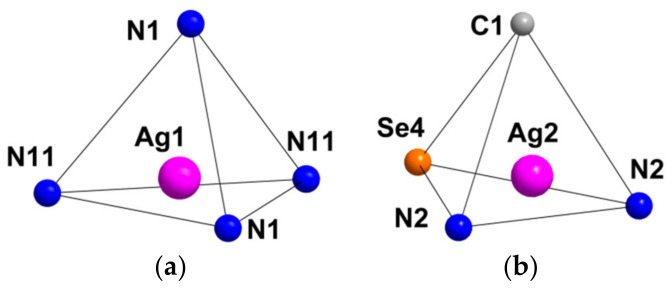
Coordination polyhedral of Ag1 (**a**) and Ag2 (**b**) atoms in the structure of **4**.

**Figure 13 molecules-27-07684-f013:**
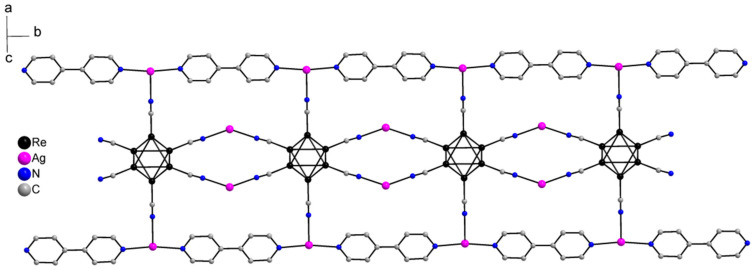
A fragment of the polymeric ribbon compound **4**.

**Figure 14 molecules-27-07684-f014:**
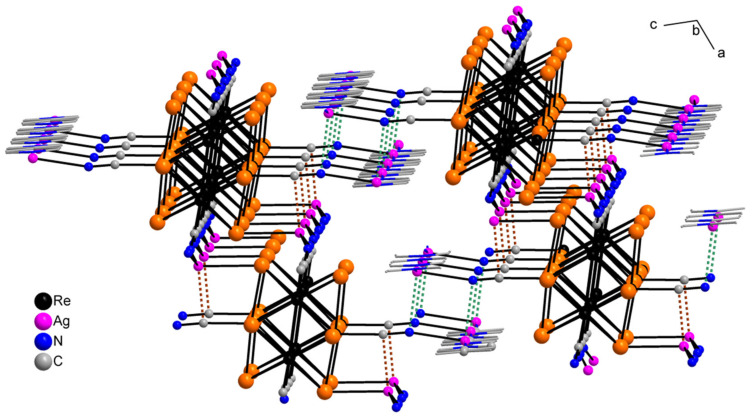
A framework fragment of compound **4**. Green dotted lines are Ag–N bonds. Red dotted lines are Ag–C bonds.

**Figure 15 molecules-27-07684-f015:**
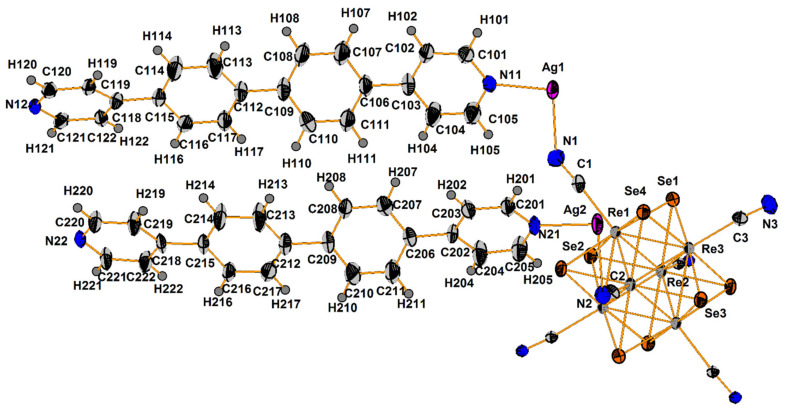
ORTEP drawing of the asymmetric unit in the structure of **5**. Thermal ellipsoids of 75% probability.

**Figure 16 molecules-27-07684-f016:**
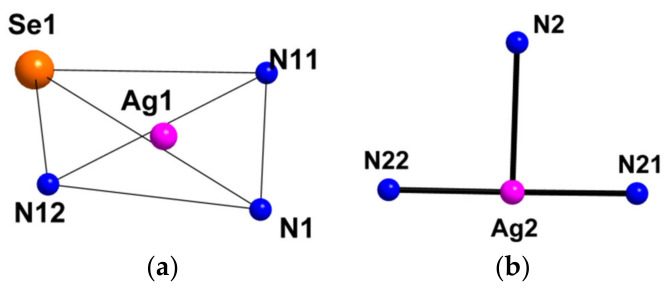
Coordination polyhedral of Ag1 (**a**) and Ag2 (**b**) atoms in the structure of compound **5**.

**Figure 17 molecules-27-07684-f017:**
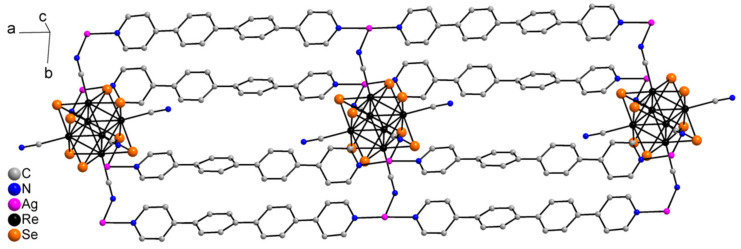
1D fragment of compound **5**.

**Figure 18 molecules-27-07684-f018:**
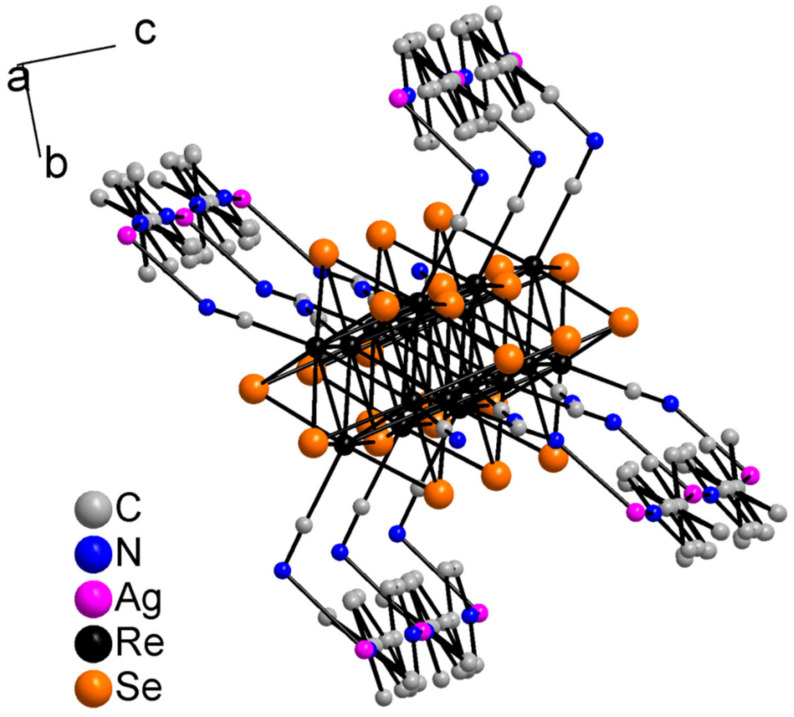
1D fragment of compound **5** along ***a*** axis.

**Figure 19 molecules-27-07684-f019:**
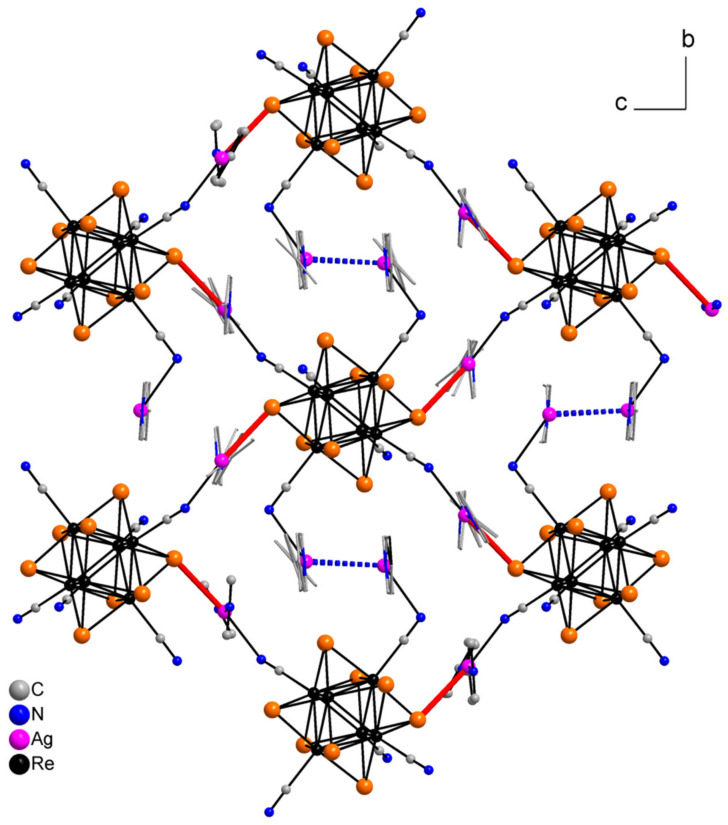
A framework fragment of compound **5**.

**Figure 20 molecules-27-07684-f020:**
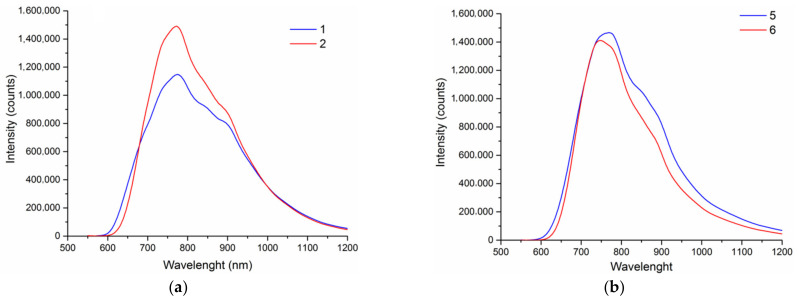
Emission spectra of: (**a**) **1** (*λ*_em_ = 775 nm) and **2** (*λ*_em_ = 770 nm); (**b**) **5** (*λ*_em_ = 762 nm) and **6** (*λ*_em_ = 746 nm).

**Figure 21 molecules-27-07684-f021:**
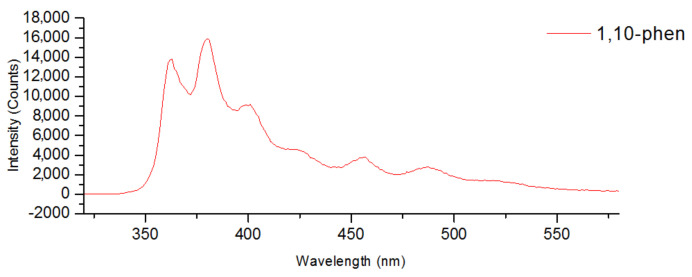
Emission spectrum of 1,10-phen.

**Figure 22 molecules-27-07684-f022:**
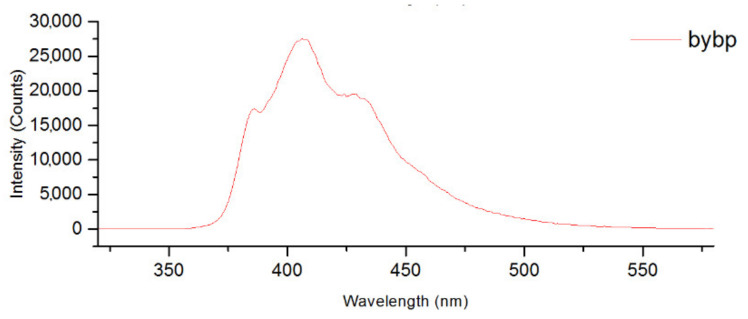
Emission spectrum of dpbp.

**Table 1 molecules-27-07684-t001:** Spectroscopic and photophysical data of the solid samples of **1**, **2**, **5** and **6** and the parent compounds.

Compound	*λ*_em_, nm	*τ*, μs	*QY*
**1**	775	1.43 (0.45), 33 (0.55)	0.05
**2**	770	0.55 (0.35), 16 (0.65)	0.06
**5**	762	9.6	0.02
**6**	746	11.4	0.04
(Bu_4_N)_4_[Re_6_S_8_(CN)_6_] [65]	746	5.85	0.021
(Bu_4_N)_4_[Re_6_Se_8_(CN)_6_] [65]	737	14.1	0.049

## Data Availability

Data are contained within the article and Appendix A.

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
