# Peer review of "Coordination Polymers Based on Rhenium Octahedral Chalcocyanide Cluster Anions and Ag+ Cations with Bipyridine Analogs"

_molecules, 2022, doi:10.3390/molecules27227684_

Round 1

Reviewer 1 Report

The manuscript by Mironov and co-workers reports the syntheses of six rhenium coordination polymers based on octahedral cluster anions and Ag cation coordinated by bipyridine derivatives. The work is the continuation of their research findings and has been further extended to luminescence application. It is an interesting manuscript and deserves to be published in Molecules after the authors address the following points.

1.       The first sentence of the abstract is misleading as the Ag cations are not only coordinated by 2,2′-bipyridine or 4,4′-bipyridine but also by 1,10-phenanthroline or bipyrimidine so better to use bipyridine derivatives in general.

2.       There is no comment on the reproducibility of the reported compounds, in particular compounds 6 with an isolated yield of only 5 mg (14%).

3.       It would be great that the authors provide powder XRD data with simulation to provide more insight into the purity of the compounds.

4.       The quality of structures seems to be very good, despite that I would recommend that the authors should provide checkcif files with the revised manuscript.

5.       In figures word clearance should be replaced by clarity.

6.       Authors must improve the English of the manuscript. The manuscript comprises of many incomplete or unclear sentences and grammatical mistakes. Some are highlighted below but the authors should improve it through out manuscript. For instance;

i) Page 1, line 30, “The current trend in the chemistry of coordination  poymers is aimed to constructing of compounds with interesting multifunctional properties, what rhenium chalcocyanide clusters are great for”

ii) Page 2, line 48, “In spite of photoactive silver compounds have been known for a long time, synthesis and investigation of compounds based on the Ag+ cations did not elaborated enough due to their low photostability”

iii) Page 2, line 74, “To determine quantum yield spectrofluorometer 74 equipped with integrating sphere Quanta-φ.”

iv) Page 7, line 224, “The asymmetric unit contains contain half of……”

iv) Please check for appropriate prepositions and helping words in the manuscript.

Reviewer 2 Report

The manuscripts report the synthesis of new heteronuclear metal complex materials based on [Re6Q8(CN)6] (Q = S or Se) clusters and Ag+ complexes with 2,2'-bipyrimidine, 2,2'-bipyridine, 4,4'-bipyridine , 1,10-phenanthroline and 4,4'-di(4-pyridyl)biphenyl. This work is a systematic development of the authors' previous studies (ref. 21). All structures are thoroughly characterized, and the influence of the nature of the organic ligand on the structure of the material is considered. Some of the materials exhibit interesting emission properties, which makes it possible to consider such materials as promising photoactive components. The manuscript is well written, provided with all necessary illustrations. There are few notes about formatting drawbacks:

line 14 - the bracket is missing after Di(4-pyridyl)biphenyl

lines 110, 128, 141, 208 - the title of should be bold.

the excitation wavelength should be noted in the title figure  20 .

there is double numeration of references.

The article can be published after minor revision.

Reviewer 3 Report

The work presented by Litvinova et al. describes the synthesis of six new coordination polymers based on the anionic clusters [Re6Q8(CN)6]4– (Q = S, Se) and cationic silver centres coordinated with bidentate nitrogen ligands. In the present case 1,10-phenanthroline, 2,2′-bipyrimidine, 4,4′-bipyridine and 4,4'-di(4-pyridyl)-1,1'-biphenyl were employed to achieve coordination polymers with diverse structural features. All the compounds were characterized by single-crystal X-ray diffraction and most of them showed weak argentophilic interactions, which are not commonly observed phenomena for most of the previously synthesized analogous compounds (especially for those bearing the 1,10-phenanthroline ligand). Also, luminescence in the red region was observed for all the compounds and lifetimes and quantum yields were remarkably higher than the ones observed for the most coordination polymers based on rhenium clusters. The bibliography appears to be appropriate, the concepts are clearly expressed and the results are properly discussed, even though several oversights can be found. Herein the observations/modifications the authors should take into consideration:

- Page 1, Title: the Authors refer to the ligands as “bypiridine derivatives”. This was done, of course, for a matter of simplification. On the other hand, 1,10-phenanthroline, although it can be conceptually derived from 2,2'-bipyridine, is a distinct heterocycle, with completely different stereo-electronic features, as well as 4,4'-di(4-pyridyl)-1,1'-biphenyl ligand. The Authors may consider changing “derivatives” to “analogues”.

- Page 1: Abstract, line 7: please, change “or 4,4′-bipyridine derivatives” to “…, 4,4′-di- and 4,4′-bipyridine derivatives”.

- Page 1, Abstract, line 16: please, change “in red region” with “in the red region”. Other similar oversights can be found within the manuscript. Please, re-check the text and replace them with the correct expressions.

- Page 1, Introduction, lines 22-23: please, insert the proper reference at the end of the first sentence. Also, what do the Authors mean with “… for two recent decades”? Were the studies interrupted at a certain point in the past? Could “for the last two decades” be more appropriate? Please, clarify.

- Page 1, Introduction, line 28: change “this fascinating properties” with “these fascinating properties”. Please, re-check the text for similar mistakes.

- Page 3, lines 110 and 128: please, change the font of the sentences in bold.

- Page 3, line 134: correct “was?”. Please, re-check the text and for similar typos.

- Page 4, line 178: “The last distance is longer probably, due to…”, please place the correct punctuation in the sentence and re-check for it throughout the whole document.

- Page 17, Conclusions, lines 432-441: this section is extremely brief. The Authors may consider adding one/two sentences about the future perspectives their new study can open for the scientific community.

- Please, change the nickname “bypb” for the ligand 4,4'-di(4-pyridyl)-1,1'-biphenyl. “Bypb” sounds misleading.

In conclusion, I believe that the work submitted by Litvinova et al. is a novel contribution to the study of rhenium chalcocyanide clusters and bipyridine-, dipyridine- and phenanthroline-based silver cationic complexes, with particular regard to the less common argentophilic interactions.

Therefore, the revised version of this work is recommended for publication on Molecules, provided the Authors will modify the manuscript accordingly.

Kind regards

Round 2

Reviewer 1 Report

I am satisfied with the the revised version of the manuscript as the authors have taken into account all the suggestions and have modified accordingly. The manuscript can be accepted in the present form, however, the "title of the supplementary information" should be modified and be the same as of the manuscript.